# The Influence of the Digital Supply Chain on Operational Performance: A Study of the Food and Beverage Industry in Indonesia

**Mohammad Agung Saryatmo \*** and **Vatcharapol Sukhotu**

School of Management, Asian Institute of Technology, Pathumthani 12120, Thailand;
vsukhotu@aggienetwork.com
* Correspondence: st119473@ait.asia

**Abstract:** In this rapidly developing digital era, digital transformations take place within every industry, and they have effects on the management of the supply chains. The aim of this study is to delve into the influence of the digital supply chain on the quality, productivity, and cost reduction aspects of operational performance. This study relies on quantitative methodology and data collected from the food and beverage industry of Indonesia. Data from a survey comprising a total of 209 responses were selected for investigation. PLS-SEM was used to perform the analysis. The investigation reveals that the digital supply chain has significant effects on operational performance in terms of quality, productivity, and cost reduction performance. This study contributes to the understanding of supply chain management by addressing the knowledge gap associated with the digital supply chain. In particular, it has concentrated on the hitherto unresearched effect of operational performance in the context of the Indonesian manufacturing industry.

**Keywords:** digital supply chain; operational performance; food and beverage industry

## 1. Introduction

Technology has advanced exponentially in recent years, with the phenomenon of digitalization being a necessity for all industries. The digitalization of supply chains is expected to grow and play a crucial role within the supply chain management (SCM) context. Numerous companies seek to be more 'digital' as a result of their own observations of the relevance and usefulness of digital technology for business development [1]. They are compelled to adapt, or they will be left behind [2]. Many companies remain unaware of how to incorporate digital technologies into their businesses. Despite the numerous advantages that digital technology provides, many companies are hesitant to invest in it properly and the majority of company revenue still comes from traditional management practices.

SCM has been on the agenda of the senior management of many industrial companies over the past decade. Scholars have also increased their attention on SCM with a focus on various aspects of the field: supplier selection [3,4], supplier involvement [5], supplier alliances [6], upstream related research in supply chain [7,8], manufacturer and retailer linkages [9], resilience of the supply chain [10], sustainability and green supply chains [11,12], and so on. However, the role of the digital supply chain has not yet been fully explored [13]. Any implementation to accomplish the target level of digitalization on the supply chain is still a complex topic and requires a clear understanding of its impact and benefits on operational performance [14]. Furthermore, as supply chain digitalization is still in its infancy in terms of development, there is still considerable room for further research in the future [15].

Indonesia, the world's largest archipelago country, comprises of over 17,000 islands and more than 580 languages, with a population of more than 270 million [16]. The Indonesian food and beverage industry was chosen for this study for a number of reasons.

Firstly, this industrial sector is a significant sector of the economy in terms of its contribution to the nation's Gross Domestic Product (GDP) in Indonesia. Statistically, as evidenced in the BPS-Statistics Indonesia [17], this manufacturing industry has made by far the largest contribution, amounting to around 36% of total GDP. Secondly, the food and beverage industry has become one of five prioritized sectors in the Making Indonesia 4.0 project, instigated by the Indonesian Ministry of Industry [18]. Making Indonesia 4.0 was announced, with key innovations like the Internet of Things (IoT), robotics, and sensor technology at its heart, with the aim of elevating Indonesia to a global Top 10 economy by 2030. Thirdly, the food and beverage industry is also the largest subsector of the Indonesian manufacturing industry, accounting for roughly a quarter of the total manufacturing value [19]. Lastly, in the Indonesian context, there is no clear evidence of the influence of the digital supply chain on operational performance for this sector. The need for research that can provide empirical evidence to support the influence of the digital supply chain on operational performance is therefore urgent, given the importance of this industry to the Indonesian economy. It is critical to find ways to strengthen the supply chain for this manufacturing sector, as these changes would be beneficial to Indonesia's economy.

This study fills the gap in past research because it focuses on the relationship between the digital supply chain and operational performance in terms of quality, productivity and cost reduction performance in the food and beverage industry of Indonesia through an empirical study. The purpose of this study is to examine the effect of the digital supply chain on operational performance, as this is a relatively new field of study [15] and there has, hitherto, been no study into the Indonesian food and beverage industry. A research question can be raised, namely, what is the relationship between the digital supply chain and operational performance? Examining these relationships is very important because it allows for a better understanding of the impact of the digital supply chain on operational performance. This paper is structured as follows: firstly, the literature around the constructs that form the basis of a theoretical model is reviewed, and a series of hypotheses is developed; then, the study's methods are explained; finally, the research findings and implications are presented and discussed.

## 2. Literature Review

### 2.1. Digital Supply Chain

Several scholars and experts have defined the meaning of 'digital supply chain'. Farahani et al. [20] define the digital supply chain as "leveraging innovative digital technologies to change the traditional way of (1) performing supply chain planning and execution tasks, (2) interacting with all kinds of supply chain participants, and (3) enabling new corporate business models". According to Büyüközkan and Göçer [15], the digital supply chain is "an intelligent best-fit technological system that is based on the capability of massive data disposal and excellent cooperation and communication for digital hardware, software, and networks to support and synchronize interaction between organizations by making services more valuable, accessible and affordable with consistent, agile and effective outcomes". Based on these definitions of the digital supply chain, it can be summarized as a new approach with innovative technology that is capable of changing the traditional supply chain operation so that improved and more efficient integration among supply chain members is achieved.

Farahani et al. [20] consider the key technological innovations that have the greatest effect on the digital supply chain to be big data, cloud computing, blockchain, IoT and robotics.

1. Big data: By providing an integrated platform for tracking performance and customer engagement through real-time data analysis and critical decision-making scenarios, big data contributes to improved visibility. As a result, the likelihood of supply chain disturbances and delays is decreased [21].
2. Cloud computing: Cloud technology allows the storage and processing of large volumes of data in real time, with information available to all supply chain partici-

pants [22,23]. In comparison to traditional information technology solutions, cloud technologies enable rapid acquisition and deployment without requiring a company to significantly extend or change its existing infrastructure [24], thus allowing the company to change as necessary.

3. Blockchain: Blockchain implementations contribute to the expansion of the complexity and size of monitoring and tracing systems [25] and also enhance the visibility and transparency of the supply chain through the use of record-keeping functions [26].

4. IoT: The real-time data generated by IoT enable the tracking of supply chain activities from product design to end user, providing reliable and timely data to assist businesses in adapting to market changes [27].

5. Robotics: Autonomous robots are expected to continue to evolve in this field in the future, enabling individuals to move to more strategic, less risky, and higher-value jobs [28]. Robotics offers manufacturers greater versatility than other types of automation [29]. Robotics also answers a question of those working in the supply chain: how will the company improve efficiency and save money? [30]. In doing so, the 21st century has witnessed many companies investing much of their revenue in adopting technology in the supply chain and a massive number are considering the investment of their money in robotics and automation [31].

### 2.2. Operational Performance

Bartezzaghi and Turco [32] stated that "operational performance comprises the actual outputs of operation strategies employed, which are influenced by operating circumstances and represent or reflect internal properties of a manufacturing system". Similarly, Lu et al. [33] state that operational performance is "a key enabler to the overall supply chain performance, which is usually the amalgamated outcome from multiple factors and enablers in the system". It is important for researchers to be more precise and explicit concerning the features of the performance measurement systems they investigate [34]. When measuring supply chain performance, a company needs to focus on financial metrics, for example cost, profitability, revenue and return on investment, and also non-financial metrics, including process quality and flexibility [35]. For a number of reasons, operational performance was selected as one of the variables. Firstly, operational performance is a major enabler of supply chain performance and receives a lot of research attention [36,37]. Secondly, operational performance is a quantifiable variable, which may be influenced by the digital supply chain. Thirdly, there is no doubt operational performance is an important and necessary component of several performance measurement systems currently in use [35,38], albeit the findings are not always consistent. Lastly, Neely [39], Ageron, et al. [40], stated that operational performance can be dependent upon quality, costs, productivity, flexibility, and dependability. In addition, based on previous studies carried out by Maani and Sluti [41], Ward and Duray [42], Wong et al. [43], and Tracey et al. [44], quality, productivity, and cost have been chosen as the performance factors.

## 3. Conceptual Framework and Hypotheses

### 3.1. Conceptual Framework

The rationale underlying this overall framework is a need to investigate the influence of the digital supply chain on operational performance, which is illustrated in the review of the literature. Most published work has investigated traditional SCM elements and their impact on operational performance [43,45–48]. These earlier studies lay the foundations for our study, which focuses on a number of innovative technologies such as big data, cloud computing, blockchain, IoT, and robotics in SCM that have been taken to describe a digital supply chain [15,49–51]. This is the first research to examine how these variables affect operational performance. The digital supply chain is comprised of more than the above five technologies. The selected five technologies are considered relevant, in terms of technology advancement to the food and beverage industry in Indonesia. Figure 1 illustrates the conceptual framework for the study.

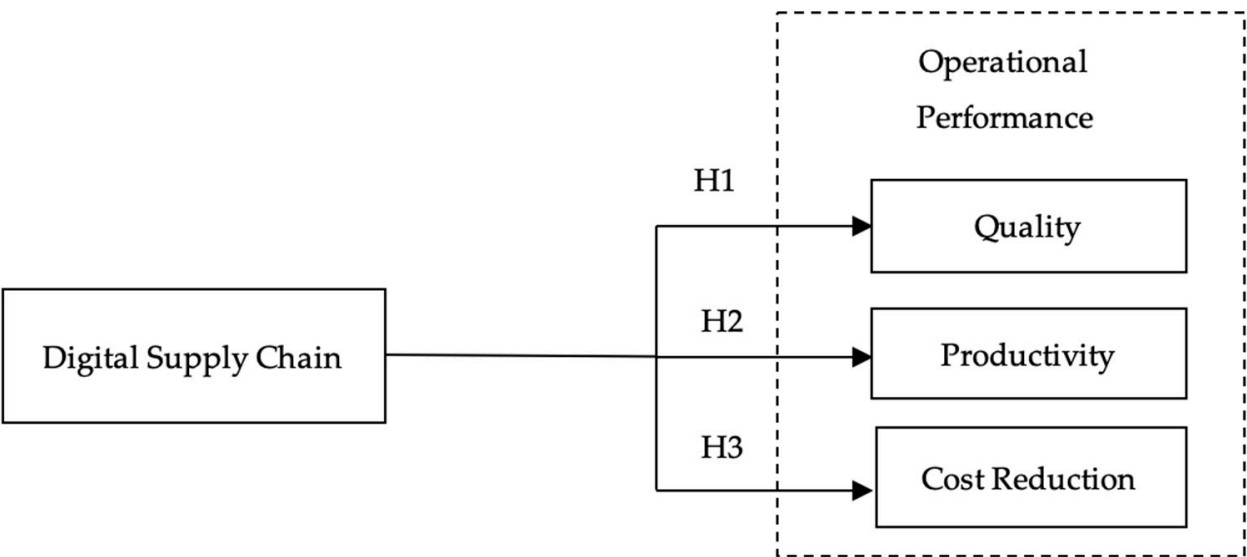

**Figure 1.** The Conceptual framework of the research.

*3.2. Hypotheses*

3.2.1. The Digital Supply Chain and Quality Performance

The first element in our operational performance is quality. In their research, Fawcett et al. [52] investigate the processes by which information technology influences supply chain performance, including product quality and inventory control. The study determined that supply chain integration and the organizing context of an information-sharing culture make the greatest contribution to differential company success. Moreover, Kwon et al. [53] have established a positive correlation between digital technology, such as big data and company performance. Their findings prove that digitalization in the company supply chain can strengthen market competitiveness and improve data quality management. In addition, the adoption of digital technologies, especially big data analytics, has been analyzed by companies, showing the positive effects on their quality performance [54]. Furthermore, the study by Agus [55] indicates that the traditional supply chain has positive associations with quality related performance, according to a survey of Malaysian manufacturing firms. The study by Boon-itt [48] of the Thai Automotive industry highlights the importance of information technology and supply chain integration on product quality. The study suggests that specific integration of information technology can enhance product quality. Recent research by Xu, Li, Chen, and Wei [23] and Wang et al. [56] states that the emerging technology of IoT in SCM contexts enables visibility and traceability of product quality. Based upon the literature researched, the following hypothesis is proposed:

**Hypothesis 1.** *The digital supply chain has a positive effect on quality performance.*

3.2.2. The Digital Supply Chain and Productivity Performance

Abdallah, Obeidat, and Aqqad [45] discovered that supply chain practices have a major positive effect on supply chain performance in terms of efficiency and effectiveness. Based on a sample comprising 104 manufacturing companies in Jordan, they recommended that manufacturing companies increase information sharing and customer integration in order to improve supply chain performance. Ittmann [57] conducted research which exposed supply chain managers to the growing value of big data and business analytics and their ability to transform and affect the supply chain industry's success. He observed that it is only in this manner that it will be possible to continue to strengthen and improve the performance of supply chains and, through this, for organizations to remain competitive.

In addition, Ellis et al. [58] emphasize that involving IoT in the chain of supply will also support the manufacturing productivity performance, including speed, flow, and quality. Moreover, since blockchain technology is increasingly considered as a next generation information tool, researchers discovered that incorporating blockchain technology into SCM activities can have an impact on supply chain relationship efficiency and growth, thereby affecting SC performance outcomes [59]. In addition, Zhou et al. [60] also find that digital technology, such as blockchain, may lead to significant improvements in productivity and efficiency. Digital technology has enormous potential, as it can improve manufacturing processes across the entire value chain [61]. Based upon the literature researched, the following hypothesis is proposed:

**Hypothesis 2.** *The digital supply chain has a positive effect on productivity performance.*

### 3.2.3. The Digital Supply Chain and Cost Reduction Performance

Previous research has demonstrated that digital technologies look for ways to support cost reduction performance. For instance, Gunasekaran and Ngai [62] have discovered that the supplier-customer integration of digital information technology systems is a cost-cutting choice for collaborative work. Moreover, data from 260 manufacturing companies were collected by Zhu and Kraemer [63], which found that digital information technologies are significantly and positively linked to company performance (cost reduction, profitability, and inventory efficiency). Additionally, Dehning, Richardson and Zmud [46] examined the financial benefits of IT investments from 123 manufacturing companies. According to their findings, supply chain implementation of IT had a substantial effect on overall financial performance. Recently, using data collected from the Chinese manufacturing industry, Yu et al. [64] discovered that a data-driven supply chain improved supply chain capabilities and related significantly to financial performance success. Furthermore, Raman et al. [65] surveyed employees of multinational companies in Asia, Australia, Europe, the Middle East, and the United States. Their findings indicate that data analytics and IoT have an effect on supply chain cost savings, consumer satisfaction, and operational excellence. In his study, Nair [66] ascertained that cloud-based technology in SCM would allow real-time pricing. Visibility of each component of the supply chain can help control costs. Additionally, incorporating blockchain technology into SCM helps improve its performance in terms of cost-efficiency, transparency, traceability, sustainability, and trust [67]. Based upon the literature researched, the following hypothesis is proposed:

**Hypothesis 3.** *The digital supply chain has a positive effect on cost reduction performance.*

### 4. Research Methodology
*4.1. Survey and Data Collection*

This study employs a quantitative methodology, as its purpose is to test a set of hypotheses. Furthermore, when examining the relationship between a variety of variables, a quantitative approach is more applicable. Within quantitative analysis, researchers typically use one of two methodologies: survey research or experimental research [68]. A survey research methodology is employed in this study, as this approach helps provide standardized information to describe variables and to examine variables and their relationships [69]. Survey-based research enables the assessment of large populations and generates data based on real-world findings (empirical data) within a comparatively brief period of time and at a reasonable cost [70,71].

A questionnaire was piloted with academics and practitioners to ensure its content validity and terminology and was then updated as necessary. Respondents to the final research questionnaire were individuals in senior and mid-level management in each company who were considered to have the appropriate knowledge of the digital supply chain and operational performance within their organization. The questionnaire was completed electronically, and the responses to the survey questions were ranked and

analyzed using a Likert scale of five points. The scale of responses for the items of the digital supply chain (independent variable) and operational performance (dependent variable) ranged from 1 to 5, where 1 signifies strongly disagree, 2 signifies disagree, 3 signifies neutral, 4 signifies agree, and 5 signifies strongly agree.

Using a simple random sampling technique, e-mails were sent to 1781 companies in the Indonesian food and beverage industry requesting completion of Google Forms. A total of 622 failed messages were received. Therefore, a total of 1159 e-mail invites were actually delivered. Seven of the 216 completed surveys had incomplete answers and were removed from the data analysis process. As a result, the total number of available survey responses received was 209, a response rate of 18%. The adequacy of this response level is supported by Sekaran and Bougie [72], where they specified that the optimal response rate for social science studies should be between 5% and 35%.

### 4.2. Analysis Technique

The framework of the current research has two group variables, digital supply chain as an independent variable and operational performance (quality, productivity, and cost reduction performance) as a dependent variable, which were measured using a questionnaire instrument. The measurement scales of the survey instruments used in this research have been taken from the literature research. Moreover, this survey was conducted in Indonesia, which therefore required a translation of the questionnaire from English to Bahasa Indonesia. The measurement items for each variable are summarized in Appendix A. This unit of analysis refers to manufacturing plants. The use of the unit of analysis adopted here follows that reported in several previous studies [32,48,73,74].

### 4.3. Measurements

To examine the research model, the PLS-SEM technique was used in conjunction with SmartPLS 3.3.3 software. This approach was applied as an analysis technique for the reasons now stated. Firstly, PLS-SEM is well suited to quantitative data analysis. PLS-SEM utilizes a bootstrapping technique to determine the importance of path coefficients [75]. Secondly, the exploratory aspect of the analysis necessitates the use of PLS-SEM [75]. Thirdly, PLS-SEM is less demanding in relation to the smallest sample size possible [76]. Lastly, PLS-SEM is considered an appropriate analytical tool where there may be concern over distribution issues, such as lack of normality [77]. According to Hair et al. [77], the data were usually distributed based on the kurtosis and skewness values (between $-1$ and $+1$). According to Appendix B, the ranges exceeded the guidelines and were non-normally distributed.

## 5. Research Results

### 5.1. Demographic Profile

Descriptive statistics (N = 209) provide information about the demographic profiles of the participants in the final survey, and these are summarized in Table 1.

<div align="center">**Table 1.** Descriptive statistics.</div>

| N = 209 | | Frequency | Percentage (%) |
|---|---|---|---|
| Age of the company | 0–10 | 59 | 28.2 |
| | >10–20 | 65 | 31.1 |
| | Over 20 | 85 | 40.7 |
| Number of employees | 0–20 person | 4 | 1.9 |
| | 20–99 person | 59 | 28.2 |
| | >=100 person | 146 | 69.9 |
| Legal entity status | Limited company (PT) | 199 | 95.2 |
| | Limited partnership (CV) | 4 | 1.9 |
| | Private/Individual company | 6 | 2.9 |
| Educational background | High school and Diploma | 27 | 12.9 |
| | Undergraduate degree | 130 | 62.2 |
| | Master's degree | 50 | 23.9 |
| | Doctoral degree | 2 | 1 |
| Years of experience in the company | 5 to 10 years | 170 | 81.3 |
| | 11 to 20 years | 29 | 13.9 |
| | Over 20 years | 10 | 4.8 |
| Role in the organization | Supervisor | 69 | 33 |
| | Department head | 11 | 5.3 |
| | Assistant manager | 8 | 3.8 |
| | Manager | 91 | 43.5 |
| | Vice director | 5 | 2.4 |
| | Director | 25 | 12 |

*5.2. Data Analysis*

5.2.1. Assessment of the Measurement Model

The aim of the measurement model is to determine the indicator loadings, internal consistency reliability, convergent validity, and discriminant validity of the observed variables (as determined by the questionnaire) in conjunction with unobserved variables [76]. Indicator loadings of more than 0.708 are suggested, since they indicate that the variable explains more than 50% of the variance in the indicator, implying sufficient item reliability [76]. According to Table 2, the indicator loadings were close to or above 0.70. However, 0.70 is usually regarded as close enough to 0.708 to be considered acceptable [77]. Cronbach's alpha was used, as well as composite reliability (CR) for internal consistency reliability. Internal consistency is assumed to be best assessed using CR rather than Cronbach's alpha as it conserves the observed variables' standardized loadings [76]. The minimum Cronbach's alpha and CR values in the PLS-SEM analysis, according to Hair et al. [75], should be greater than 0.70. Cronbach's alpha and CR values were all greater than 0.70, as shown in Table 2, showing that all the indicators of each variable have satisfactory internal consistency reliability. The Average Variance Extracted (AVE) of each latent variable was calculated to ensure that the variables were convergently correct [76]. When the AVE is 0.50 or higher, it indicates that the variable explains at least 50% of the variance of its items [77]. According to Table 2, all variables have an AVE greater than 0.50. In other words, the AVE values of each variable lie within the required range (>0.50). This implies that the AVE values for each variable meet the acceptance criteria and convergent validity is adequate for the measurement model.

**Table 2.** Construct reliability and validity.

| Main Variables | Items | Loadings | Cronbach's Alpha | CR | AVE |
|---|---|---|---|---|---|
| Digital Supply Chain (DSC) | DSC_1 | 0.865 | 0.934 | 0.944 | 0.630 |
| | DSC_2 | 0.803 | | | |
| | DSC_3 | 0.803 | | | |
| | DSC_4 | 0.762 | | | |
| | DSC_5 | 0.876 | | | |
| | DSC_6 | 0.873 | | | |
| | DSC_7 | 0.723 | | | |
| | DSC_8 | 0.717 | | | |
| | DSC_9 | 0.793 | | | |
| | DSC_10 | 0.702 | | | |
| Quality Performance (QP) | QP_1 | 0.738 | 0.769 | 0.852 | 0.591 |
| | QP_2 | 0.781 | | | |
| | QP_3 | 0.819 | | | |
| | QP_4 | 0.733 | | | |
| Productivity Performance (PP) | PP_1 | 0.763 | 0.777 | 0.857 | 0.601 |
| | PP_2 | 0.841 | | | |
| | PP_3 | 0.777 | | | |
| | PP_4 | 0.714 | | | |
| Cost Reduction Performance (CP) | CP_1 | 0.751 | 0.786 | 0.862 | 0.609 |
| | CP_2 | 0.814 | | | |
| | CP_3 | 0.811 | | | |
| | CP_4 | 0.744 | | | |

The discriminant validity of the latent variables was the next measurement. This determines how empirically distinct a variable is from the other variables in the structural model. Discriminant validity refers to the degree to which different variables' measurements vary from one another. The discriminant validity of PLS-SEM is evaluated by using two measures: (1) the Fornell–Larcker criterion test and (2) cross loadings. The Fornell–Larcker criterion was used to compare the model's squared correlations to the correlations of other latent variables, as shown in Table 3. All AVEs have much greater square root values than all other cross correlations, and the AVE values of each variable exceed 0.50. Overall, discriminant validity for this measurement model can be accepted, and it supports the discriminant validity between variables.

**Table 3.** Fornell–Larcker criterion test.

| | CP | DSC | PP | QP |
|---|---|---|---|---|
| **CP** | 0.781 | | | |
| **DSC** | 0.621 | 0.794 | | |
| **PP** | 0.511 | 0.573 | 0.775 | |
| **QP** | 0.553 | 0.679 | 0.564 | 0.769 |

Examining the cross loadings of the indicators of the scales used in testing the study model is another technique for determining discriminant validity [78]. The cross loadings are calculated by correlating each latent variable's component scores with each of the indicator variables [79]. As shown in Table 4, the cross-loading of all observed variables was greater than each variable's inter-correlations with all other observed variables in the model. Furthermore, these findings corroborated the standards for assessing cross-loadings and provided acceptable validation for the measurement model's discriminant validity. Consequently, the proposed conceptual model was demonstrated to be acceptable, with adequate reliability, convergent validity, discriminant validity, and validation of the research model confirmed.

**Table 4.** Cross-loadings test.

| Main Variables | DSC | QP | PP | CP |
|---|---|---|---|---|
| DSC_1 | **0.865** | 0.576 | 0.480 | 0.540 |
| DSC_2 | **0.803** | 0.527 | 0.440 | 0.521 |
| DSC_3 | **0.803** | 0.505 | 0.447 | 0.502 |
| DSC_4 | **0.762** | 0.548 | 0.468 | 0.436 |
| DSC_5 | **0.876** | 0.603 | 0.462 | 0.521 |
| DSC_6 | **0.873** | 0.559 | 0.480 | 0.527 |
| DSC_7 | **0.723** | 0.477 | 0.448 | 0.431 |
| DSC_8 | **0.717** | 0.551 | 0.386 | 0.458 |
| DSC_9 | **0.793** | 0.480 | 0.449 | 0.508 |
| DSC_10 | **0.702** | 0.554 | 0.484 | 0.475 |
| QP_1 | 0.469 | **0.738** | 0.424 | 0.437 |
| QP_2 | 0.495 | **0.781** | 0.427 | 0.361 |
| QP_3 | 0.610 | **0.819** | 0.389 | 0.462 |
| QP_4 | 0.498 | **0.733** | 0.508 | 0.437 |
| PP_1 | 0.401 | 0.446 | **0.763** | 0.355 |
| PP_2 | 0.494 | 0.473 | **0.841** | 0.453 |
| PP_3 | 0.466 | 0.466 | **0.777** | 0.430 |
| PP_4 | 0.408 | 0.357 | **0.714** | 0.332 |
| CP_1 | 0.482 | 0.402 | 0.401 | **0.751** |
| CP_2 | 0.514 | 0.443 | 0.481 | **0.814** |
| CP_3 | 0.500 | 0.454 | 0.335 | **0.811** |
| CP_4 | 0.440 | 0.427 | 0.376 | **0.744** |

### 5.2.2. Assessment of the Structural Model

Following the evaluation of the measurement model, the structural model was assessed. Hair et al. [76] recommend examining the coefficient of determination ($R^2$), the predictive relevance ($Q^2$), and structural model path coefficients. $R^2$ measures the explained variance of a latent variable relative to its total variance. The greater the $R^2$, the better the independent latent variable's ability to explain the dependent latent variable. $R^2$ values of 0.67, 0.33, and 0.19, respectively, can be considered "substantial," "moderate," and "weak." [78]. However, Hair et al. [76] note that appropriate $R^2$ values vary according to the model's complexity and the study discipline. Given the degree to which these phenomena are already well known, one would anticipate a relatively high $R^2$. A lower $R^2$ is suitable for less well-understood phenomena [80]. $R^2$ values as low as 0.10 are considered satisfactory in certain disciplines, for example, when forecasting stock returns [81]. Thus, because the study of the digital supply chain is in its initial stages [15], a lower range of $R^2$ is acceptable. Table 5 presents the results of the $R^2$ coefficients which are shown schematically in Figure 2.

**Table 5.** $R^2$ coefficients.

| Variable | $R^2$ |
|---|---|
| Quality Performance | 0.461 |
| Productivity Performance | 0.329 |
| Cost Reduction Performance | 0.386 |

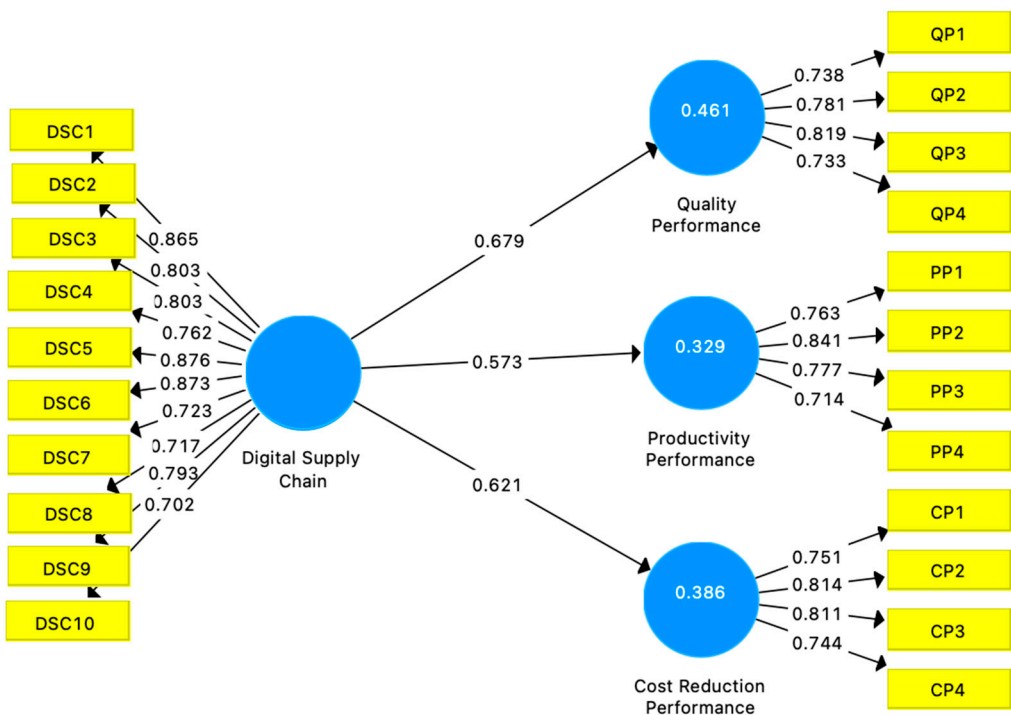

**Figure 2.** The Coefficient of determination ($R^2$).

The predictive relevance ($Q^2$) technique measures the SmartPLS model's quality, and it is estimated by using the procedure of blindfolding. In this study, cross-validated redundancy was performed. Accordingly, Chin [78] states that when $Q^2$ values are greater than zero, the model is assumed to have predictive relevance. Figure 3 summarizes the $Q^2$ values for the model, showing a value at 0.260 for quality performance, 0.188 for productivity performance, and 0.231 for cost reduction performance, all of which are higher than the threshold limit.

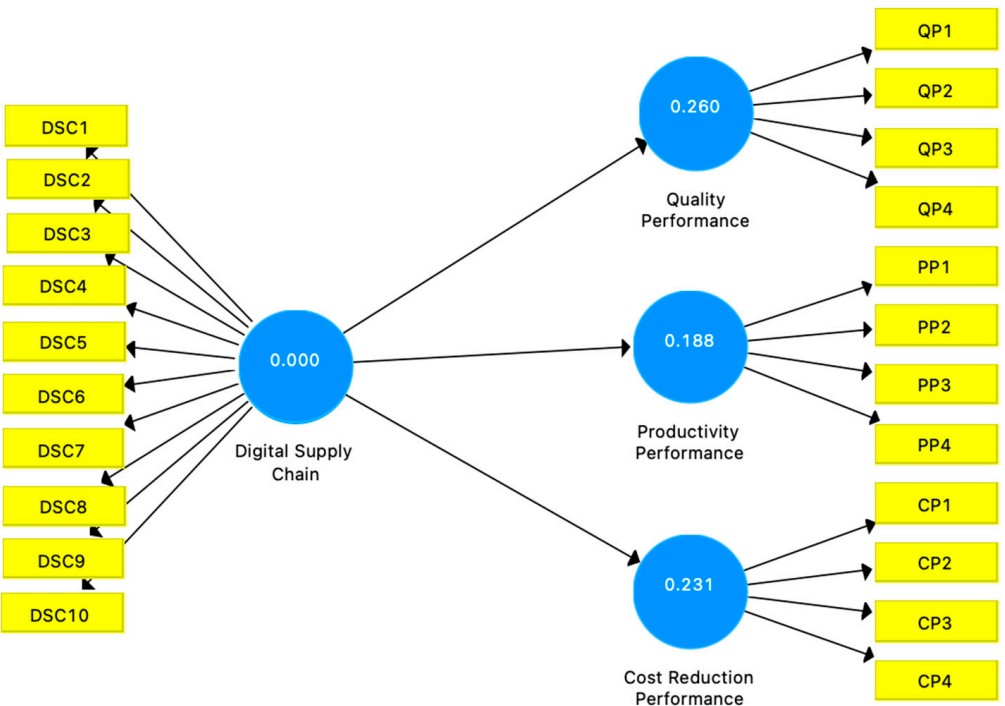

**Figure 3.** Predictive relevance of the model ($Q^2$).

Structural path coefficients allow this study to confirm or refute each hypothesis and better understand the degree to which dependent and independent variables are related. To determine the significance of the hypothesis, the bootstrapping procedure was used [76]. To determine the significance of the path coefficient and T-statistics values, a bootstrapping procedure was performed using 5000 subsamples with no sign changes, as detailed in Table 6 and shown schematically in Figure 4. The results in Table 6 confirm that the digital supply chain positively influences the quality performance ($\beta = 0.045$, T = 15.099, $p < 0.000$). As a result, H1 is supported. Additionally, the results also support H2 and H3. There is a positive influence from the digital supply chain on productivity performance ($\beta = 0.057$, T = 10.114, $p < 0.000$) and cost reduction performance ($\beta = 0.063$, T = 9.928, $p < 0.000$).

**Table 6.** Path coefficients.

| Hypothesis | Relationship | T-Value | P-Value | Decision |
|---|---|---|---|---|
| Hypothesis 1 | DSC → QP | 15.099 | 0.000 | Supported |
| Hypothesis 2 | DSC → PP | 10.114 | 0.000 | Supported |
| Hypothesis 3 | DSC → CP | 9.928 | 0.000 | Supported |

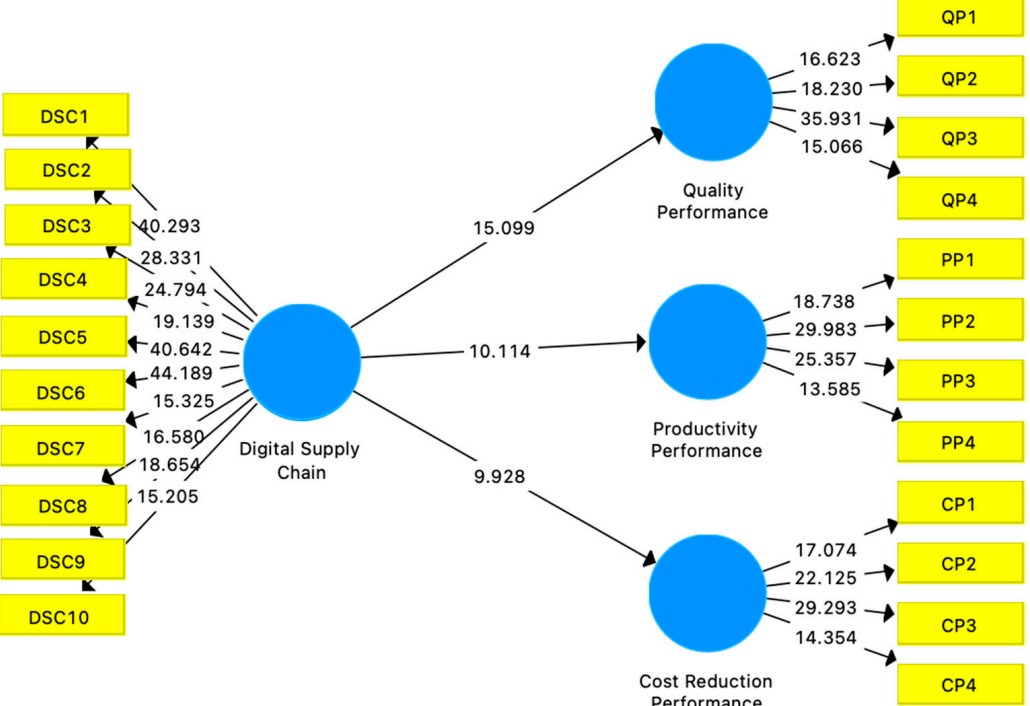

**Figure 4.** The structural path coefficients.

## 6. Discussion

This empirical study analyzes the relevant theoretical framework in the food and beverage industry of Indonesia. The findings indicate that the digital supply chain has a beneficial effect on quality performance. The finding is consistent with previous research. For example, Li and Wang [82] recommended that companies in the food industry investigate digital innovation opportunities derived from big data and develop appropriate data-driven strategies to enhance their product quality and market competitiveness. Furthermore, the finding also supports those by Fawcett et al. [52] who showed how digital technology could be used to enhance supply chain performance and create the capability to share information. Companies that recognize the need for, and invest in, digital supply chains should achieve higher levels of performance such as product quality, inventory, and supply chain cost. The findings from the work presented here are consistent with the results

of previous studies carried out by Raguseo [83]. He found that digital technology in terms of big data technology has a positive effect on company performance. Big data is widely recognized as a critical field of future technology and is rapidly gaining the attention of many industries due to the high value it can offer businesses. Additionally, the finding here is congruent with that of Brandyberry et al. [84] who found that by using information technology, companies could manage the flow and impact of various supply chain dimensions, such as quality, flexibility, cost, and delivery, by leveraging information technology.

In addition, the results indicate that the digital supply chain has a positive influence on productivity performance. This result is in line with previous research undertaken by Pilat and Criscuolo [85], who discovered that emerging innovations have the potential to boost productivity by encouraging innovation and lowering the costs of a variety of business processes. Despite the rapid growth of digital technologies, they stated that industries also face particular challenges in the adoption and effective use of digital technologies, particularly in the case of productivity-enhancing applications. Furthermore, digital technology creates new growth opportunities for businesses and aids them in making strategic decisions that increase their productivity [86]. Similar findings have been discovered by Ellis et al. [58], who pointed out that the digital technology such as IoT and cloud computing can capture vital data in analytics to drive end-to-end supply chain improvements. Therefore, investment, adoption, and usage of digital technologies in the supply chain alone are not enough to improve productivity performance without having optimum knowledge and information sharing systems in place.

The results also show the positive influence that the digital supply chain has on cost reduction performance. This finding corroborates previous research outcomes. For example, Maulina and Natakusumah [87] found that information technology improved operational performance in the supply chain context in relation to cost performance, responsiveness, and reliability. Similar findings have been found by Rai et al. [88], who reported that investment in IT infrastructure in the supply chain created more efficient functionalities than traditional ones, which can lead to improved performance, particularly in financial flow (revenue growth) and operational excellence. In addition, Oh and Jeong [89] noted that in order to achieve superior supply chain efficiency in terms of cost performance, companies must integrate on multiple levels, including integration on both internal and external levels, integration of functions and regions, integration of supply chains and networks, and integration of IT.

In summary, the findings of this study indicate that the digital supply chain positively influences all three aspects of operational performance (quality, productivity, and cost reduction). The implementation of digital supply chain practices in a company leads to an increase in the level of operational performance. The adoption of digital technology can create considerable value-added and monetary gain for companies, and it will soon become a standard throughout the industry. Companies have to consider the importance of selecting digital technologies such as big data, cloud computing, blockchain, IoT, and robotics for their supply chain. The findings are also supported by previous research from Haddud and Khare [90], who highlighted the importance of companies identifying possible areas for improvement and ensuring that all potential supply chain digitalization benefits are fully realized. Therefore, the adoption of digitalization in the supply chain might then be seen as being incremental rather than radical [91]. On the other hand, since digital technology adoption is both complex and time-consuming, companies must possess specific implementation skills and an understanding of their objectives.

## 7. Conclusions

The purpose of this study was to determine how the digital supply chain affects operational performance. A conceptual framework and a research methodology were developed to investigate the impact of the digital supply chain on operational performance. Then, a quantitative method of research was designed, and data were collected via questionnaires distributed to respondents, focusing on the Indonesian food and beverage industry. Three

hypotheses of the proposed model are supported. There is a statistically significant correlation between digital supply chain performance and operational performance (in terms of quality, productivity, and cost reduction performance). This research focuses exclusively on one sector, the food and beverage industry in Indonesia. As a result, extending the study to include countries with varying cultural traditions and different levels of new technology adoption may offer an opportunity for future research. Furthermore, in order to further develop this area, research may be carried out in other chosen industries.

## 8. Managerial Implications

Several managerial implications flow from the findings of this research. Firstly, this study raises knowledge and comprehension of the meagre prior research on the relationship between the digital supply chain and operational performance. The conceptual framework introduced in this research can be used to assist managers in the manufacturing industry in acquiring a deeper comprehension of the effects of the digital supply chain on operational performance. Secondly, empirical evidence indicates that the digital supply chain results in increased operational performance. These findings will help managers acquire a greater understanding of the factors that affect the output of digital supply chains, especially in developing countries with similar characteristics. Furthermore, digital supply chain investment necessitates in-depth research, and supply chain parameters may need to be reconfigured and redefined as a result. Thus, managers should be knowledgeable about the different forms of emerging technology that might be worth investing in.

## 9. Limitations and Recommendations for Future Research

As with any research, constraints on the scope of this study offer opportunities for further exploration. Firstly, there is a limitation related to the context of the study, manufacturers classified under the food and beverage industry in a single country (Indonesia). Consequently, the results cannot be generalized to economies in less developed or more developed stages or to other industries. Thus, future research should replicate this approach for countries with different levels of technology development, as well as in other industry sectors such as the agricultural industry. Secondly, data collection was conducted entirely by self-report questionnaires. Despite criticism from some researchers, this approach was deemed appropriate due to the difficulties associated with independently assessing each of these variables. Thus, in the future, research may be applied to cross-sectional data comparing organizational output prior to, and following, the adoption of the digital supply chain concept by manufacturing industry companies.

**Author Contributions:** All authors contributed equally to the research presented in this paper and to the preparation of the final manuscript. All authors have read and agreed to the published version of the manuscript.

**Funding:** This research received no external funding.

**Institutional Review Board Statement:** Not applicable.

**Informed Consent Statement:** Not applicable.

**Data Availability Statement:** Not applicable.

**Conflicts of Interest:** The authors declare no conflict of interest.

## Appendix A. The Measurements

| Main Variables | Items | Statement | References |
|---|---|---|---|
| Digital Supply Chain | DSC_1 | Big data is used to improve our data quality. | Raman et al. [65]; Schoenherr et al. [92]; Cegielski et al. [93]; Ben-Daya et al. [94]; Merlino and Sproǵe [30]. |
| | DSC_2 | Our company is able to monitor customer interaction through real time data analysis. | |
| | DSC_3 | Our company is able to achieve information exchange with cloud computing. | |
| | DSC_4 | Cloud technologies enhance process capability and local storage. | |
| | DSC_5 | Blockchain improves traceability of products in the supply chain. | |
| | DSC_6 | Exchange of information with customers and suppliers is easier through the application of blockchain. | |
| | DSC_7 | IoT provides a link between customers and the company. | |
| | DSC_8 | IoT provides the linkage for all devices to the internet associated with production processes. | |
| | DSC_9 | Robotics is used to improve production capacity. | |
| | DSC_10 | Our company uses or plans to use robotics on a regular basis in the future. | |
| Quality performance | QP_1 | Our company is able to produce consistent quality products with a low rate of defects. | Maani and Sluti [41]; Safizadeh et al. [95]; Tracey et al. [44]; Koufteros et al. [96]. |
| | QP_2 | Our company operates regular customer satisfaction surveys to monitor our product quality. | |
| | QP_3 | Our company is able to maintain a low number of customer complaints concerning product quality. | |
| | QP_4 | Our company is able to supply products based on conformance quality (national and international standards). | |
| Productivity performance | PP_1 | Our labor and machine productivity is performing better than in its intended function. | Ward and Duray [42]; Wong et al. [43]. |
| | PP_2 | Our company is able to optimize our production defect/waste to acceptable levels. | |
| | PP_3 | Our company is able to provide short delivery times acceptable to our customers. | |
| | PP_4 | Our company is able to increase capacity utilization in our production when demand requires it. | |
| Cost reduction performance | CP_1 | Our company is able to manufacture products at competitive prices while maintaining a profitable operational performance. | Davis and Schul [97]; Maani and Sluti [41]; Tracey et al. [44]. |
| | CP_2 | Our company is able to produce products from a low inventory of raw materials thereby minimizing production costs. | |
| | CP_3 | Overall, our logistics costs (including distribution, transportation, and handling costs) can be reduced year on year through our supply chain management. | |
| | CP_4 | The reductions in cost achieved are considerably better value than expected. | |

**Appendix B. Calculation of the Mean, Standard Deviation, Excess Kurtosis, and Skewness**

|         | Mean  | Standard Deviation | Excess Kurtosis | Skewness |
|---------|-------|--------------------|-----------------|----------|
| DSC_1   | 3.943 | 0.889              | −0.303          | −0.503   |
| DSC_2   | 4.115 | 0.873              | 0.16            | −0.791   |
| DSC_3   | 4.077 | 0.861              | −0.07           | −0.648   |
| DSC_4   | 4.177 | 0.765              | 0.496           | −0.701   |
| DSC_5   | 3.962 | 0.89               | 0.096           | −0.621   |
| DSC_6   | 3.962 | 0.874              | −0.195          | −0.531   |
| DSC_7   | 4.244 | 0.734              | −0.15           | −0.639   |
| DSC_8   | 4.191 | 0.74               | 0.901           | −0.752   |
| DSC_9   | 4.115 | 0.816              | 1.764           | −1.013   |
| DSC_10  | 4.033 | 0.899              | 0.694           | −0.863   |
| QP_1    | 4.167 | 0.761              | 0.991           | −0.817   |
| QP_2    | 4.349 | 0.69               | 0.112           | −0.765   |
| QP_3    | 4.239 | 0.726              | 0.965           | −0.78    |
| QP_4    | 4.411 | 0.687              | −0.127          | −0.837   |
| PP_1    | 4.206 | 0.765              | −0.239          | −0.628   |
| PP_2    | 4.287 | 0.766              | 1.108           | −0.992   |
| PP_3    | 4.306 | 0.665              | −0.274          | −0.539   |
| PP_4    | 4.321 | 0.73               | −0.258          | −0.731   |
| CP_1    | 4.182 | 0.78               | 0.979           | −0.88    |
| CP_2    | 4.201 | 0.731              | 1.071           | −0.778   |
| CP_3    | 4.167 | 0.822              | 0.524           | −0.841   |
| CP_4    | 4.278 | 0.764              | −0.043          | −0.778   |

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
