# Peer review of "The Influence of the Digital Supply Chain on Operational Performance: A Study of the Food and Beverage Industry in Indonesia"

_sustainability, doi:10.3390/su13095109_

Round 1
Reviewer 1 Report
The paper investigates the impact of digital supply chain on operational performance (i.e. quality, productivity, and cost reduction) in the context of food and beverage industry in Indonesia. The paper is well organize. The analysis part for research methodology is well explained. However there are several issues about research framework and methodology to be addressed for further improvement.
- The authors provide several references about the influences of digital supply chain (DSC) and digital transformation (DT) on performance. Actually, most of them point out DSC and DT enable some factors (e.g. collaboration, partnership, information sharing, risk reduction, etc.) to improve the performance. In this paper, there is no further research advancement. In addition, there is no research foundation or theory for the research framework proposed.
- The definition of operation performance has serious problem. Are the factors quality, productivity, and cost reduction independent? Is there is any collaboration among them? Will that be any problem for the study?
- It is also not clear about what level is for the constructs (or variables) in the study. Are they referring to organization level and supply chain (inter-organizational level)? Is the operation performance is for the focal company?
- Most of the measurement is based on the self-report and perception. Though the authors also point out this issue in the limitation, it is indeed is a serious weakness for the study.
- The definitions of digital supply chain is quite vague. Is it referring to the adoption of the technologies as mentioned in the paper? Are those technologies to comment with external parties in supply chain or mainly for internal improvement? Is digital supply chain same as digital transformation? Or digital transformation is more appropriate?
- What are the qualification and requirements for the subjects in the paper? How to ensure they meet the requirement (e.g. knowledge of those technology, awareness of the adoption, and understanding of the company performance)?
- It is not clear why not to include other types of performance factors.
Other issues
- The definitions of operational performance from reference [32] and [33] are almost identical (“usually is the amalgamated outcome from multiple factors and enablers in the system” (p. 3). Please remove the redundancy.
- In the literature review, it is suggested to point out the research methodology (e.g. empirical study, modeling, case study) adopted.
- “Moreover, a quantitative methodology is more applicable when examining the relationship among a variety of variables is required. Furthermore, when examining the relationship between a variety of variables, a quantitative approach is more applicable” (p.5). Are these sentences are the same?
- “A survey research methodology is employed in this study, as this approach helps provide standardized information to describe variables and to examine variables and their relationships” (p.5). Please explain the reason why (instead of what) to choose survey as the research methodology.
Reviewer 2 Report
This manuscript focuses on the technological side of digital supply chains and its impact on their operational performance.
My main concern relates to the way hypotheses are defined, based on the literature review. It is not so clear if, for instance, references supporting hypothesis 1 really focus on quality performance or performance as a whole in all its components. It is important to be more specific while defining this hypothesis, so that the reader can be sure it is derived from information that only focuses on the quality component of performance. The same comment applies for the two other hypotheses.
My second comment concerns the three components of performance investigated, namely quality, productivity and cost-reduction. How do you capture the trade-off between each component? How can you be completely certain that what is measured is only the effect you are looking at? There might be noise coming from the multi-component performance assessment in your analysis. It may be an explanation of your very low R².
My last comment is devoted to the various components of performance investigated. As cost reduction performance increases when the cost is reduced, while for productivity, performance increases when productivity improves. How do you handle this double direction in your evaluation?
Lines 200-202: the same sentence is duplicated.
Reviewer 3 Report
Article present a quantitative research aiming to determine the relation between how companies implement digital supply management and the operational performance. It is a good article, clearly presented, with a research question that is supported by the research. The methodology is also described in great detail.
Just two comments. At line 226 it is mentioned that the research has two variables, while the rest of the article suggests more.
Also there is a leap of faith in the article`s story. Authors define digital supply chain by enumerating some technologies (big data, cloud computing, iot ...) then responders respond and relate those technologies to their operational performance then the authors relate DSC with performance. In the same logic they could have linked directly IoT + Big Data + Blockchain with operational performance.
Round 2
Reviewer 1 Report
The manuscript was revised according to the comments in the review report. The revision is satisfactory.
Reviewer 2 Report
Thank you for considering my comments in your new manuscript.
Regarding hypothesis 1 : the last reference 'Aymerich et al [24]' refers to productivity and efficiency that are related to hypothesis 2 and to cost-effectiveness that is related to hypothesis 3.
Could you provide of a description of H1 that just focuses on quality ? Or do the 3 indicators so closely related to each other that it is a factor that cannot be qualified independently of the two others ?
Otherwise, I do not have additional comments
